# Planning deficits in Huntington's disease: A brain structural correlation by voxel-based morphometry

Jesus Calderon-Villalon[1], Gabriel Ramirez-Garcia[2], Juan Fernandez-Ruiz[2,3], Fernanda Sangri-Gil[1], Aurelio Campos-Romo[4], Victor Galvez[1,4] *

**1** Laboratorio de Neurociencias Cognitivas y Desarrollo, Escuela de Psicología, Universidad Panamericana, Ciudad de México, México, **2** Laboratorio de Neuropsicología, Departamento de Fisiología, Facultad de Medicina, Universidad Nacional Autónoma de México, Ciudad de México, México, **3** Instituto de Neuroetología, Universidad Veracruzana, Ciudad de México, México, **4** Unidad Periférica de Neurociencias, Facultad de Medicina, Instituto Nacional de Neurología y Neurocirugía "MVS", Universidad Nacional Autónoma de México, Ciudad de México, México

* vgalvez@up.edu.mx

**Data Availability Statement:** We uploaded the tables with the planning scores and their correlations with gray matter in figshare with the DOI: https://doi.org/10.6084/m9.figshare.

## Abstract

### Introduction

Early Huntington's disease (HD) patients begin to show planning deficits even before motor alterations start to manifest. Generally, planning ability is associated with the functioning of anterior brain areas such as the medial prefrontal cortex. However, early HD neuropathology involves significant atrophy in the occipital and parietal cortex, suggesting that more posterior regions could also be involved in these planning deficits.

### Objective

To identify brain regions associated with planning deficits in HD patients at an early clinical stage.

### Materials and methods

Twenty-two HD-subjects genetically confirmed with incipient clinical manifestation and twenty healthy subjects were recruited. All participants underwent MRI T1 image acquisition as well as testing in the Stockings of Cambridge (SOC) task to measure planning ability. First, group comparison of SOC measures were performed. Then, correlation voxel-based morphometry analyses were done between gray matter degeneration and SOC performance in the HD group.

### Results

*Accuracy* and *efficiency* planning scores correlated with gray matter density in right lingual gyrus, middle temporal gyrus, anterior cingulate gyrus, and paracingulate gyrus.

13672873.v3. However, the individual patients' data is confidential, and we are prevented from publicly sharing these data. For more information, contact: Víctor Gálvez, PhD Cognitive Neuroscience Laboratory, Psychology, Health Sciences School, Universidad Panamericana, CDMX city, México. Telephone: +52 1 55 54 82 16 00 ext. 6429 e-mail: vgalvez@up.edu.mx.

**Funding:** This study was supported in part by CONACYT grant No. 220871 and No. A1-S-10669, PAPIIT-UNAM grant No. IN220019 to JFR, and CONACYT fellowship No. 574022/403010 to GRG. V. Gálvez received a grant "Fondo semilla 2019" from FCS-Universidad Panamericana.

**Competing interests:** The authors have declared that no competing interests exist.

## Conclusions

Our results suggest that planning deficits exhibited by early HD-subjects are related to occipital and temporal cortical degeneration in addition to the frontal areas deterioration.

## Introduction

Huntington's Disease (HD) subjects is a neurodegenerative disease characterized by motor, behavioral, and cognitive deficits observed usually during middle adulthood (35–44 years old). These changes are associated with an abnormal expansion of the CAG trinucleotide repeat sequence in the gene 4p 1.3 [1]. Early changes that may precede clinical manifestation include executive function deficits on flexibility and planning that reduce their goal-directed behaviors in daily life [2]. Planning ability is defined as the identification and organization of the elements needed to carry out an intention or to achieve a goal [3]. This ability does not depend on a single brain region but has been related to a set of regions associated with the fronto-cingulo-limbic-parietal network [4].

The Tower of London (TOL) task has been used as a reliable measure of planning ability [5–10]. Functional imaging analyses have found that the basal ganglia, the premotor cortex, and the dorsolateral prefrontal cortex are the main regions activated during its execution [5–7]. However, previous studies with patients have revealed that subjects with posterior cortical damage also show impairments in its global execution [8, 9]. Additionally, it has been shown that the functional activity of the dorsal ("where") and ventral ("what") visual pathways correlate with the execution but no with the complexity of the task, suggesting that these posterior areas may have a sensory involvement during the visuospatial planning process, in contrast to the more anterior structures [9, 10].

Early HD neurodegeneration pattern also involve a decrease of gray matter (GM) in occipital, parietal, and motor cortices, in addition to the prefrontal cortex atrophy [11], suggesting that regions associated with planning abilities are broadly affected in this illness. This is supported by reports describing planning alterations in manifest HD-subjects, as a consequence of functional decoupling of the medial prefrontal cortex (mPFC) and the left premotor area [2, 12].

To delve into the neural bases of the planning deficits in HD, here we investigated the possible relation between the neurodegeneration of specific brain areas and planning deficits observed in early HD-patients. For this purpose, GM from whole-brain structural T1 magnetic resonance imaging (MRI) was correlated with the performance on the Stockings of Cambridge (SOC) planning task. We hypothesized that planning deficits exhibited by early HD-subjects are related to the cortical degeneration in both posterior and frontal brain regions.

## Materials and methods

### Participants

Twenty-two HD gene-mutation carriers at the early clinical stage and twenty healthy subjects matched for age, sex, and level of education participated in this study (Table 1). The HD-subjects were recruited at the Instituto Nacional de Neurología y Neurocirugía (INNN). After the positive molecular diagnosis, the patients were invited to participate in the study. Subsequently, the cognitive assessment was performed at the INNN and the brain imaging acquisition was performed at the Instituto Nacional de Psiquiatría. The inclusion criteria for the HD-

**Table 1. Clinic and demographic groups data.**

| | Control group | HD-subjects |
|---|---|---|
| Men:Women | 7:13 | 9:13 |
| Age (years) | 45.4 ± 12.2 (26.5–67.5) | 46.1± 12.1 (27.6–67.5) |
| Education (years) | 16.1 ± 2.8 (10–21) | 14.13 ± 3.2 (9–19) |
| Age onset (years) | - | 45.3 ± 10. 6 (26.8–62. 6) |
| Symptoms duration (months) | - | 48.5 ± 51.0 (0–186.2) |
| CAG-repeat length | - | 45.3 ± 3.8 (40–54) |
| Disease burden* | - | 389.9 ± 100.5 (154.8–534) |
| TFC | - | 11.9 ± 1.8 (8–13) |
| UHDRS motor scale | - | ± 13.6 (0–40) |

*Disease burden was calculated using the following formula: Age (years) * (CAG-repeat length– 35.5) [22]. ±
standard deviation.

subjects were a positive molecular genetic diagnosis for HD and a Total Functional Capacity
score (TFC) higher than 7 points. The exclusion criteria was the presence of neuropathological
findings in the MRI non-HD related. The control group subjects self-reported no history of
neurological or psychiatric disorders. They were recruited at the same period of evaluation as
the HD-subject group by an open invitation to the general public. All the procedures were per-
formed according to the Declaration of Helsinki [13], approved by the health and ethics com-
mittees of the INNN and the Universidad Nacional Autónoma de México (UNAM) (N˚ DIC/
419/14 and N˚ 41/14). All participants signed a written informed consent before their inclu-
sion in this study.

## Clinical and neuropsychological testing

MoCA [14] was used to evaluate the global cognitive status for all subjects. The TFC [15] and
the motor subscales from Unified Huntington's Disease Rating Scale (UHDRS) [16] were used
to measure the clinical status of the HD group. Particularly, the symptomatic HD patients
were defined by the scores obtained in the TFC scale, where a lower score reflects a reduced
functional capacity. The TFC range is from 0 to 13 points, consequently, in our study we
defined as early symptomatic HD to the subjects with scores higher than 7, considering the
scale criteria [15].

**Planning performance task with SOC.** Planning performance in all participants was
evaluated using the SOC test, a digital task from the Cambridge Neuropsychological Test
Automated Battery (CANTAB) software version Eclipse®. It requires spatial abilities and stra-
tegic planning and is aimed to give a measure of frontal lobe function [17, 18]. Participants
performed the task using a computer tablet; the touch screen showed two panels. The upper
panel showed the reference pattern that the subject had to match in the lower panel. The objec-
tive was to move three colored balls, placed in different positions within three stockings to the
reference position showed in the upper panel. The participants could only move the balls one
at a time by selecting the required ball, then selecting the position to which it should be
moved. The balls were arranged in different patterns in each problem, and participants were
instructed to make as few moves as possible to solve it. The level of difficulty increased as the
number of minimum moves needed to complete the task was risen. Twelve problems were
evaluated. The first six problems could be solved in a minimum of two movements (and four
at the most). The last six problems were considered "difficult" because the number of mini-
mum moves needed to solve them was five (and twelve at the most). Two types of

measurements were collected: the *accuracy* (the sum of the number of solved problems in the minimum of movements allowed, being 12 for all problems and 6 for the "difficult" problems), and *efficiency* (the sum of the number of movements made in the respective set of problems: all and "difficult" problems) [19]. It should be noted that in contrast to accuracy where a higher score is better, in efficiency a higher count indicates worse performance.

## Image acquisition

All images were acquired using a 3T MRI scanner (Philips Medical Systems, Eindhoven, The Netherlands). The high-resolution anatomical acquisition consisted of a T1-3D Fast Field-Echo sequence with the following parameters: TR/TE: 8/3.7 ms; FOV: $256 \times 256$ mm$^2$; Flip angle = 8˚; acquisition and reconstruction matrix: $256 \times 256$; and isometric resolution: $1 \times 1 \times 1$ mm$^3$.

**Voxel-based morphometry (VBM).** GM measurement was performed using VBM as implemented in FSL software (http://www.fmrib.ox.ac.uk/fsl) following the standard procedure reported previously [20]. Then, to test if there was an association between GM density and SOC measures in HD-subjects, a one-sample t-test was performed in a voxel-wise analysis through the GLM, including GM and *accuracy* and *efficiency* scores in all problems (12 evaluated problems) and "difficult" problems (last 6 evaluated problems). The significance level was set at $p < .01$, and the Family Wise Error correction for multiple comparisons was done using the random permutation method ($n = 10,000$) using Threshold-Free Cluster Enhancement (TFCE), suggested for general linear model inference analysis [21]. For all the analyses, the disease burden score (calculated using the formula: age (years) $\times$ [CAG repeat length $-$ 35.5]) [22] was included as a nuisance variable. Only clusters with a minimum cluster size of 30 voxels were reported. Coordinates were reported in the MNI standard-space and anatomical labels were obtained from the Talairach Daemon labels, MNI cerebellum, and Harvard-Oxford Cortical Structural and Subcortical Structural Atlases.

## Statistical analysis

SOC performance comparison between groups was conducted using the nonparametric Mann-Whitney U-test. *Accuracy* and *efficiency* measures were calculated for the 12 problems together, and for the 6 problems considered as "difficult". SOC scores were correlated with the patients' GM density. Spearman's rho correlation analysis was performed using the average GM density from each VBM significant cluster. Finally, only significant correlations corrected by the Bonferroni method were selected (.01/number of significant clusters obtained by VBM), setting a significance level for all problems at $p < .0016$ (.01/6), and for "difficult" problems at $p < .0025$ (.01/4). All statistical analyses were performed using SPSS software (SPSS version 23, Chicago, Illinois, USA).

## Results

### Clinical testing

The TFC scale mean ($\bar{x}$) and ($\pm$) standard deviation score from HD-subjects was $\bar{x} = 11.9 \pm 1.8$, detecting eighteen patients (82%) in Stage I (scores from 11 to 13) and four patients (18%) in Stage II (scores from 7 to 10). The mean UHDRS score was $\bar{x} = 17.1 \pm 13.6$ out of a maximum of 124. Five patients (22.7%) showed abnormal movements, with the highest score of 40 points in the most affected patient. Seventeen patients (77.3%) scored less than 17 points, and two patients (9.5%) scored 0 points. The above data confirmed their early clinical status according to the functional and motor decline [15, 23]. MoCA global score comparison

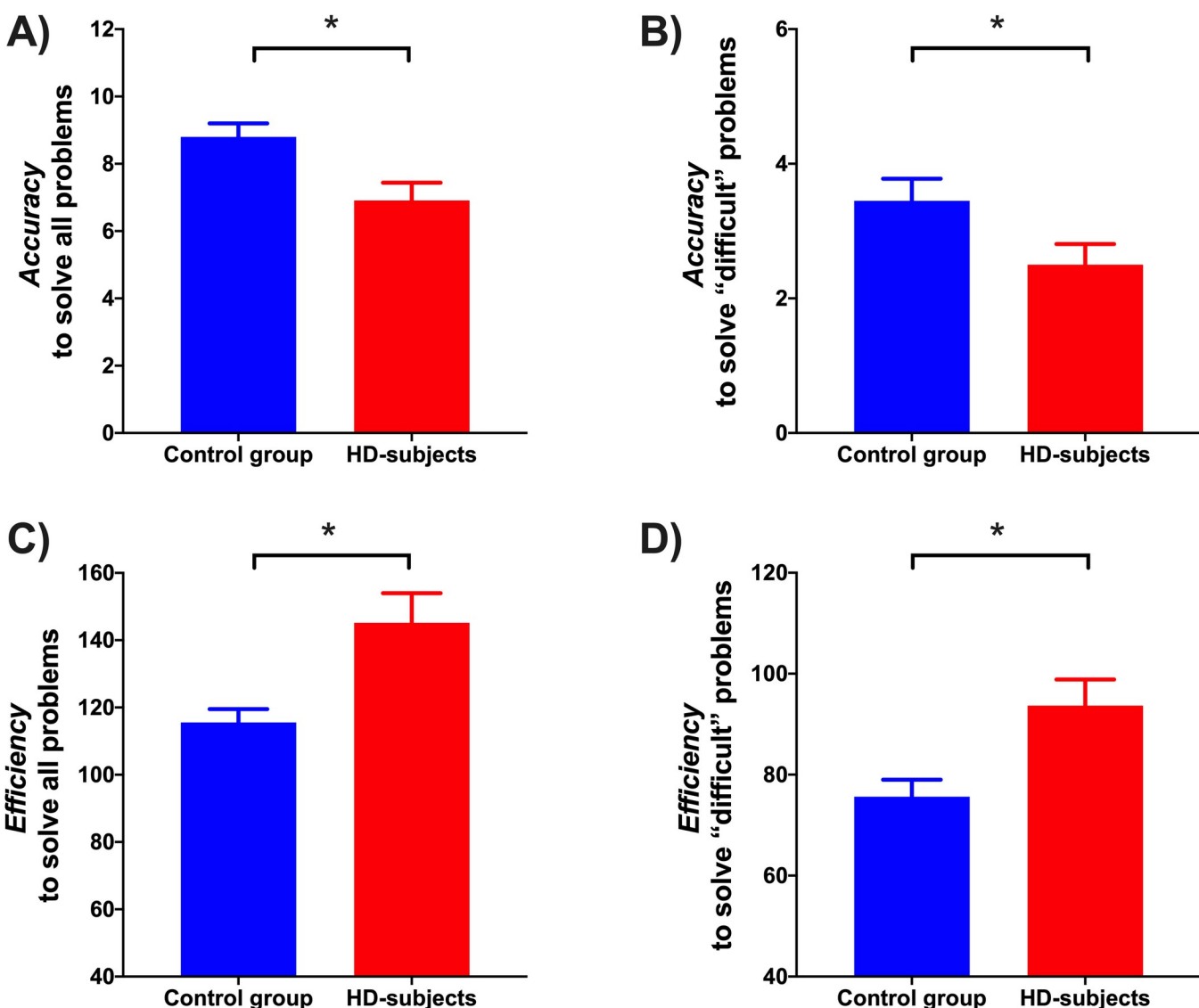

**Fig 1. Differences in SOC scores analyses between groups.** A) Accuracy to solve all problems: Number of problems solved in the minimum number of movements. B) Accuracy to solve "difficult" problems: Number of problems solved in the minimum number of movements. C) Efficiency to solve all problems: Sum of movements made to solve each of all problems. D) Efficiency to solve "difficult" problems: Sum of movements made to solve each "difficult" problem. *p < .05.

showed significant group differences between the control ($\bar{x}$ = 27.5 ± 2.2) and HD ($\bar{x}$ = 24.6 ± 2.9) groups ($U$ = 109.500, $p$ = 0.002).

SOC scores analyses also showed significant *accuracy* and *efficiency* differences between groups (Fig 1). The control group ($\bar{x}$ = 8.8 ± 1.7) showed better *accuracy* than the HD group while solving all the problems ($\bar{x}$ = 6.9 ± 2.4) (U = 117.5, p = .009). Likewise, the control group ($\bar{x}$ = 3.4 ± 1.4) showed better *accuracy* than the HD group while solving the "difficult" problems ($\bar{x}$ = 2.5 ± 1.4) (U = 139, p = .038). Regarding the *efficiency* to solve all problems, the control group showed better performance ($\bar{x}$ = 115.5 ± 17.7) than the HD group ($\bar{x}$ = 145.1 ± 41.2) (U = 101.5, p = .003). Similarly, the control group ($\bar{x}$ = 75.6 ± 15) showed better *efficiency* solving the "difficult" problems than the HD group ($\bar{x}$ = 93.6 ± 24.2) (U = 116.5, p = .009).

### Relationship of SOC scores and GM density in early manifest HD-subjects

VBM analyses showed a positive correlation between the *accuracy* to solve all problems with volume preservation in the right middle temporal gyrus (RMTG) (posterior division), right lingual gyrus (RLG), right paracingulate gyrus (RPCG), left putamen, left central opercular cortex and left insular cortex. A similar analysis found a negative correlation between the *efficiency* to solve "difficult" problems with volume preservation in the right cerebellum posterior lobe, left cerebellar nodule, RMTG (anterior division), and right anterior cingulate gyrus (RACG) (Fig 2). No significant correlations were found in the *efficiency* to solve all problems nor the *accuracy* to solve "difficult" problems.

## Discussion

Here we tested if planning deficits in HD-subjects could be related to cortical degeneration in frontal and posterior brain regions. Our analyses of the VBM suggest that besides the contribution of the frontal lobe deterioration to these deficits, posterior degeneration also correlates with the HD group's impaired performance in the planning task. Following is a discussion of these results.

### Planning performance in early HD-subjects

Early HD-subjects exhibited planning deficits associated with *accuracy* and *efficiency* performance. HD patients made significantly more movements to solve the SOC problems, suggesting impaired planning *efficiency* compared with the control group. Additionally, HD-subjects were less accurate as suggested by the smaller number of solved problems using the fewest possible moves. These SOC results confirm what has been reported in other studies employing similar tasks to explore planning skills in HD-patients [2, 12, 24].

### SOC performance association with early HD-subjects brain

Our results suggest that, in addition to frontal areas such as the cingulate gyrus, HD planning deficits are also associated with the degeneration in occipital and temporal cortices. Volume reductions in three cortical areas were correlated positively with SOC scores in the HD group: RMTG, RLG, and the cingulate gyrus (RACG and RPCG). It is important to note that VBM results showed a consistent GM density negative correlation between RMTG and cingulate gyrus with the *efficiency* to solve "difficult" problems, and with the *accuracy* to solve all SOC problems.

The possible explanation for the strong association between the *accuracy* to solve all problems and RMTG degeneration could be related to its role in visual processing, including visuospatial information processing [25, 26]. The hypothesis that planning deficits may involve visuospatial deterioration is further supported by the correlation between *accuracy* deficits with the degeneration of posterior brain areas like RLG. This is supported by previous reports showing significant activity changes in this area in Parkinson's disease patients in comparison to control subjects while performing the TOL task [27]. This is consistent with the evidence that visual perceptual deficits could give rise to changes in cognitive performance, particularly when planning ability is involved [28].

Although planning ability in HD-subjects has been associated with mPFC functional connectivity [12], our GM structural analysis showed relative preservation of this area in our patient cohort. In contrast, our analysis found a significant correlation between the RACG with low *efficiency* planning, and RPCG with low *accuracy* planning; both limbic brain regions. This may be explained by considering a previous functional MRI study in healthy subjects

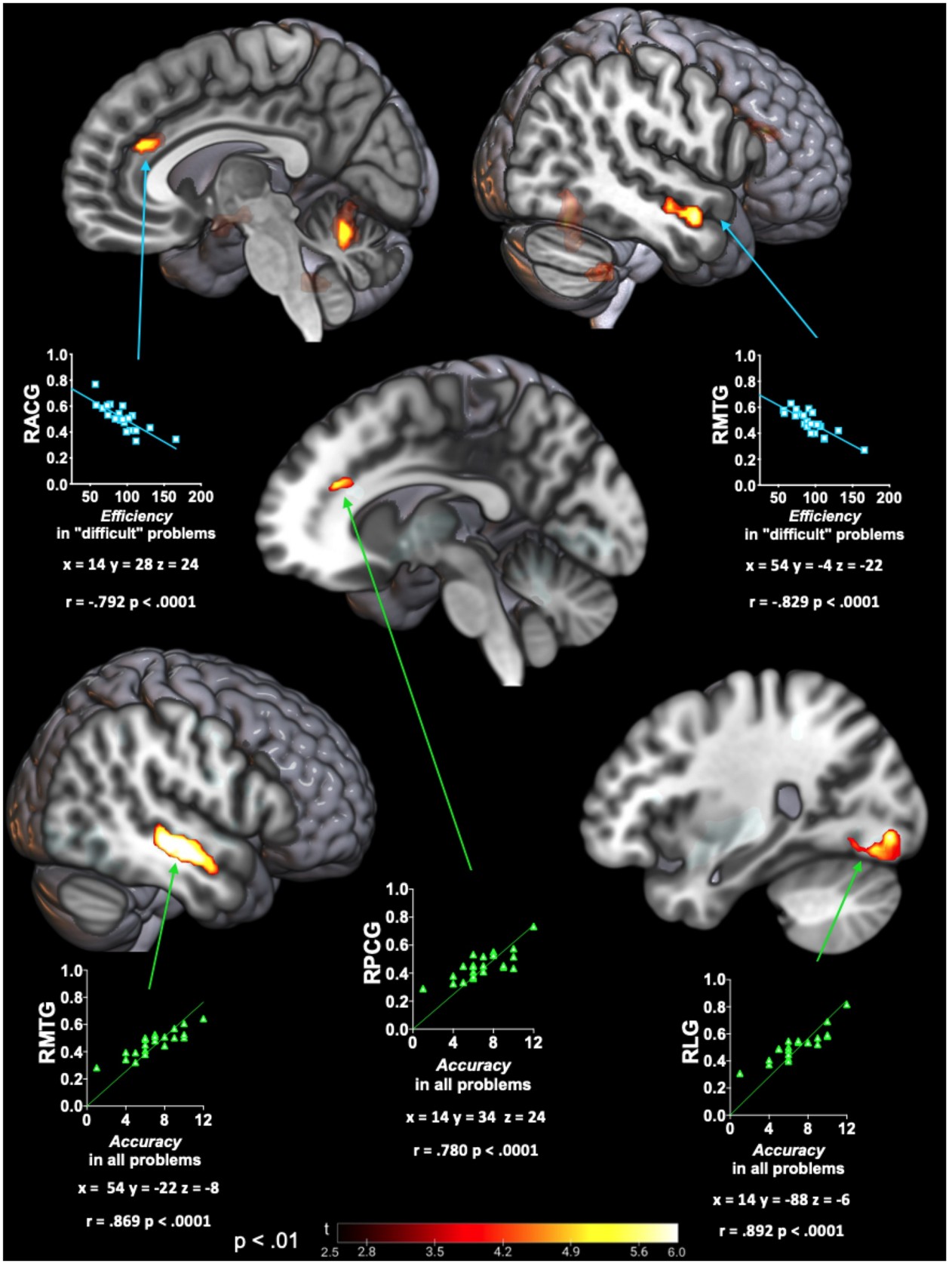

**Fig 2. VBM analysis association between GM and SOC scores.** Significant association between GM with *accuracy* to solve all problems (green scatterplot); and *efficiency* to solve "difficult" problems (blue scatterplot). Color maps indicate t-value level corrected by TFCE (p < .01). Scatter plots show the Spearman's rho correlation between the average GM density from VBM significative clusters with the corresponding SOC score. Statistical parametric maps are shown onto the three-dimensionally rendered MNI template brain. RACG: right anterior cingulate gyrus; RMTG: right middle temporal gyrus; RLG: right lingual gyrus; RPCG: Right paracingulate gyrus.

during TOL performance [29]. That study describes the prefrontal-cingulate connectivity; however, it also shows more activity in the anterior cingulate cortex to solve low difficult problems but switching to more activity in prefrontal regions to solve the more difficult problems, suggesting that both areas are differentially activated depending on the degree of cognitive demand [29]. The above could help explain the differences in brain regions that correlated with planning deficits depending on the problem complexity in HD-subjects.

In addition, RACG is reported as part of the executive attention network related to error detection (self-monitoring) [30] and arousal increase during complex problems performance [9]. In this regard, the low *efficiency* of patients may also be part of an attentional domain deficit that matches with the idea that planning ability is related to the fronto-cingulo-limbic-parietal network where different brain regions converge to coordinate executive functions [4].

Finally, the role of the subcortical structures like the putamen in planning deficits in HD-subjects should also be taken into account. The striatum is part of a wide fronto-executive network involved in planning processing, which network includes the medial prefrontal areas and the dorsolateral prefrontal areas [2, 12]. Usually, cognitive impairments exhibited in HD patients are associated to caudate nucleus decrease due to the spiny neurons atrophy [31]. This decrease seems to affect particularly the caudate nucleus during the clinical debut, specially compromising the prefrontal cortico-striatal loop [2, 32, 33]. However, our study found that putamen deterioration in HD patients also contributes to the accuracy decrease to solve planning problems. This result suggests that, in addition to the caudate nucleus, the putamen degeneration could also compromise cognitive functions during the early clinical stage. This decrease seems to affect particularly the caudate nucleus during the clinical debut, specially compromising the prefrontal cortico-striatal loop [2, 32, 33]. However, our study found that putamen deterioration in HD patients also contributes to the accuracy decrease to solve planning problems. This result suggests that, in addition to the caudate nucleus, the putamen degeneration could also compromise cognitive functions during the early clinical stage.

Some limitations in our study that should be considered for future research: 1) Because we had access to a relatively small number of patients, given the rarity of this disease, we recommend including the assessment of planning deficits in HD studies with a larger sample size. This could confirm the consistency of the brain gray matter areas associated with SOC deficits in HD reported in this study. 2) The cohort consisted of HD patients at an early symptomatic stage; it remains to be determined if prodromal patients start showing planning deficits before the clinical manifestation of the disease. 3) In this study we used two tests to analyze the cognitive impairment in general and the planning deficits in particular. It is recommended that for future assessments of the global cognitive status, and the planning ability, more neuropsychological batteries should be used, allowing a more comprehensive evaluation in these patients. 4) It would be interesting to explore if the correlation between the planning performance deficits and the loss of gray matter density could also be found with white matter deterioration, or changes in functional connectivity in HD-subjects. 5) Finally, even though we identified temporo-occipital degeneration associated with planning performance, we must be cautious with the interpretation of this result. The reported correlations passed the multiple comparison corrections, and each correlation showed different coefficients of determination. However, they

could still have a collinearity problem, which is a main challenge to overcome in this type of study.

## Acknowledgments

We thank the HD patients and healthy controls who participated in this study.

## Author Contributions

**Conceptualization:** Jesus Calderon-Villalon, Gabriel Ramirez-Garcia, Juan Fernandez-Ruiz, Victor Galvez.

**Data curation:** Jesus Calderon-Villalon, Victor Galvez.

**Formal analysis:** Jesus Calderon-Villalon, Gabriel Ramirez-Garcia, Victor Galvez.

**Funding acquisition:** Victor Galvez.

**Investigation:** Jesus Calderon-Villalon, Victor Galvez.

**Methodology:** Jesus Calderon-Villalon, Gabriel Ramirez-Garcia, Juan Fernandez-Ruiz, Victor Galvez.

**Project administration:** Victor Galvez.

**Resources:** Victor Galvez.

**Software:** Jesus Calderon-Villalon, Victor Galvez.

**Supervision:** Gabriel Ramirez-Garcia, Juan Fernandez-Ruiz, Victor Galvez.

**Validation:** Juan Fernandez-Ruiz, Victor Galvez.

**Visualization:** Victor Galvez.

**Writing – original draft:** Jesus Calderon-Villalon, Gabriel Ramirez-Garcia, Juan Fernandez-Ruiz, Fernanda Sangri-Gil, Victor Galvez.

**Writing – review & editing:** Jesus Calderon-Villalon, Gabriel Ramirez-Garcia, Juan Fernandez-Ruiz, Fernanda Sangri-Gil, Aurelio Campos-Romo, Victor Galvez.

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
