## [Decision Letter · Decision Letter 0]

13 Jan 2021

PONE-D-20-37811

Planning deficits in Huntington's disease: a brain structural correlation by voxel-based morphometry

PLOS ONE

Dear Dr. Gálvez Zúñiga,

Thank you for submitting your manuscript to PLOS ONE. After careful consideration, we feel that it has merit but does not fully meet PLOS ONE’s publication criteria as it currently stands. Therefore, we invite you to submit a revised version of the manuscript that addresses the points raised during the review process.

One expert Reviewer and myself reviewed the manuscript. You will find some points raised by me, and the others in the review from Reviewer 1. Please revise your manuscript along these lines and it should be acceptable for publication.

We look forward to receiving your revised manuscript.

Kind regards,

Andre Aleman, PhD

Academic Editor

PLOS ONE

Journal Requirements:

2. Please ensure you have discussed any potential limitations of your study in the Discussion, including study design, sample size and/or potential confounders.

3. In your Methods section, please provide additional information about the participant recruitment method and the demographic details of your participants.

Please ensure you have provided sufficient details to replicate the analyses such as:

a) the recruitment date range (month and year),

b) a description of any inclusion/exclusion criteria that were applied to participant recruitment,

c) a description of how participants were recruited, and

d) descriptions of where participants were recruited and where the research took place.

4. Please provide a sample size and power calculation in the Methods, or discuss the reasons for not performing one before study initiation.

Additional Editor Comments:

This is an interesting paper describing the results of an investigation into structural brain correlates of planning ability in patients with Huntington's Disease. The study was well-conducted and the manuscript is well written.

Some points deserve attention:

1. The neuropsychological evaluation was rather limited in scope (only MoCA and SOC). This should be acknowledged as a limitation in the Discussion.

2. The MoCA showed differences between groups in general cognitive ability. The authors could consider taking this into account as a covariate (maybe MoCA without executive component), so as to have a more "pure" estimate of planning ability as measured with SOC in association with the VBM measure.

3. The relevance of striatum is briefly mentioned in the Discussion, but as frontostriatal involvement has been regarded to be crucial for planning performance, the authors should devote a few more sentences to the question how this relates to their results.

Reviewers' comments:

Reviewer's Responses to Questions

**Comments to the Author**

1. Is the manuscript technically sound, and do the data support the conclusions?

Reviewer #1: Yes

2. Has the statistical analysis been performed appropriately and rigorously? 

Reviewer #1: Yes

3. Have the authors made all data underlying the findings in their manuscript fully available?

Reviewer #1: Yes

4. Is the manuscript presented in an intelligible fashion and written in standard English?

Reviewer #1: Yes

5. Review Comments to the Author

Reviewer #1: This paper addresses an interesting issue in HD research. Although traditionally thought of as a disease primarily associated with striato-frontal degeneration, since the TRACK-HD study (Tabrizi, 2009) we know that degeneration of posterior regions is part of early HD disease evolution. And so the question arises which cognitive and behavioural defect are associated with or attributable to posterior degeneration. In this paper the authors report on such an association.

They used one neuropsychological test, SOC, and related this to grey matter density decrease in a whole brain VBM approach. Significant differences in SOC test performance between early HD subjects and controls were found in terms of accuracy and efficiency for both “all problems” and “difficult problems”. They report (lines 199-205) a significant association between the ‘accuracy to solve all problems with volume preservation in the right middle temporal gyrus (RMTG, posterior division), right lingual gyrus (RLG), right paracingulate gyrus (RPCG), left putamen, left central opercular cortex and left insular cortex. A similar analysis found a negative correlation between the efficiency to solve “difficult” problems with volume preservation in the right cerebellum posterior lobe, left cerebellar nodule, RMTG (anterior division), and right anterior cingulate gyrus (RACG).’

(Confusingly, the next sentence (lines 205-207) states: ‘No significant associations were found on the accuracy to solve all problems neither on the efficiency to solve “difficult” problems.’ What do they mean? What do I miss or misunderstand?)

The association between SOC test performance and loss of gray matter density appears to vindicate a role in early HD for the degeneration of posterior parts in what have traditionally been considered ‘frontal executive functions’. But the problem that should be recognized in this type of analysis is the widespread degeneration that may take place and the resulting collinearity of data with resulting overinterpretation of correlations. It would be nice if this issue would be addressed in the Discussion.

Yet, I consider this an interesting paper, worth sharing with the rest of the scientific community.

Some minor issues:

In the materials and Methods section, first line, 22 HD mutation carriers and 20 healthy controls are mentioned. But Table 1 of Results, a control group of 22 (men:women = 9:13) is mentioned.

How were symptomatic patients defined and selected? As having specific motor signs (the international definition) or of having early cognitive, behavioural or functional problems prior to the onset of motor signs? I assume the latter. The ‘Clinical testing’ section mentions only 5 patients with ‘abnormal movements’ and 2 patients with UHDRS motor score of 0. The authors should explain their patient selection in more detail.

In the Discussion the authors write: ‘our analysis found a significant correlation between the RACG with low efficiency planning, and RPCG with low accuracy planning; both frontal brain regions.’ (Lines 258 and 259) I guess they mean: both limbic regions?

6. PLOS authors have the option to publish the peer review history of their article (what does this mean?). If published, this will include your full peer review and any attached files.

Reviewer #1: No

---

## [Author Response · Author response to Decision Letter 0]

17 Feb 2021

Following is a point by point response to the concerns raised by the academic editor and the expert reviewer: 

1. “Please ensure that your manuscript meets PLOS ONE's style requirements, including those for file naming.” Answer: 

We have double-checked the PLOS ONE’s style requirements to ensure that this manuscript meets the publication's requirements.

2. “Please ensure you have discussed any potential limitations of your study in the Discussion, including study design, sample size and/or potential confounders.” Answer: 

Our research presents potential limitations which we have now included in the Discussion section as follow (lines 281 – 299):

Some limitations in our study that should be considered for future research:

1) Because we had access to a relatively small number of patients, given the rarity of this disease, we recommend including the assessment of planning deficits in HD studies with a larger sample size. This could confirm the consistency of the brain gray matter areas associated with SOC deficits in HD reported in this study. 

2) The cohort consisted of HD patients at an early symptomatic stage; it remains to be determined if prodromal patients start showing planning deficits before the clinical manifestation of the disease. 

3) In this study we used two tests to analyze the cognitive impairment in general and the planning deficits in particular. It is recommended that for future assessments of the global cognitive status, and the planning ability, more neuropsychological batteries should be used, allowing a more comprehensive evaluation in these patients.

4) It would be interesting to explore if the correlation between the planning performance deficits and the loss of gray matter density could also be found with white matter deterioration, or changes in functional connectivity in HD-subjects.

5) Finally, even though we identified temporo-occipital degeneration associated with planning performance, we must be cautious with the interpretation of this result. The reported correlations passed the multiple comparison corrections, and each correlation showed different coefficients of determination. However, they could still have a collinearity problem, which is a main challenge to overcome in this type of study.

3. “In your Methods section, please provide additional information about the participant recruitment method and the demographic details of your participants.”

Answer:

We have added more information about the participants in the Methods section as follows:

Twenty-two HD gene-mutation carriers at the early clinical stage and twenty healthy subjects matched for age, sex, and level of education participated in this study (Table 1). The HD-subjects were recruited at the Instituto Nacional de Neurología y Neurocirugía (INNN). After the positive molecular diagnosis, the patients were invited to participate in the study. Subsequently, the cognitive assessment was performed at the INNN and the brain imaging acquisition was performed at the Instituto Nacional de Psiquiatría. The inclusion criteria for the HD-subjects were a positive molecular genetic diagnosis for HD and a Total Functional Capacity score (TFC) higher than 7 points. The exclusion criteria was the presence of neuropathological findings in the MRI non-HD related. The control group subjects self-reported no history of neurological or psychiatric disorders. They were recruited at the same period of evaluation as the HD-subject group by an open invitation to the general public. All the procedures were performed according to the Declaration of Helsinki (13), approved by the health and ethics committees of the INNN and the Universidad Nacional Autónoma de México (UNAM) (Nº DIC/419/14 and Nº 41/14). All participants signed a written informed consent before their inclusion in this study. 

(lines 82-96). 

4. “Please provide a sample size and power calculation in the Methods, or discuss the reasons for not performing one before study initiation.”

Answer:

One problem while studying rare diseases like HD is that there is a small number of patients. Therefore, the typical sample size and power calculations for larger populations are not easily suitable to study this type of diseases. In these instances, it has been suggested to test the whole population (Morris, 2021). The number of patients who participated in the genetic counselling program organized by the Instituto Nacional de Neurología y Neurocirugía was of 27 during the recruitment period, and only 22 completed the inclusion criteria. However, it has been suggested that a minimum sample of 12 subjects is enough to get an 80% power in similar voxel-based studies (Desmond & Glover, 2002). In addition, it has been shown that a small sample (less than 16 participants by group) with a Family Wise Error correction, may result in under-report of brain abnormalities, but there are no substantial changes in the number of them (only an increase of 2% if 10 more subjects were added) if the sample considers at least 16 subjects per group (Fusar-Poli et al, 2014). Similar findings have been suggested in other imaging fields, where it has been recommended testing between 16 and 32 subjects (Friston 2012). This suggested that our sample size of 22 subjects support this kind of analysis.

Desmond, J. E. & Glover, G. H. (2002). Estimating sample size in functional MRI (fMRI) neuroimaging studies: Statistical power analysis. Journal of Neuroscience Method, 118, 115-128.

Fusar-Poli, P., Radua, J, Frascarelli, M., Mechelli, A., Borgwardt, S., Di Fabio, F., Biondi, M., loannidis, J. P. A. & David, S. P. (2014). Evidence of reporting biases in voxel-based morphometry (VBM) studies of psychiatric and neurological disorders. Human Brain Mapping, 35, 3052-3065. 

Friston, K. (2012). Ten ironic rules for non-statistical reviewers. Neuroimage, 61(4), 1300-1310.

Morris, E. (2004). Sampling from small populations. Retrieved from http://uregina.ca/~morrisev/Sociology/Sampling%20from% 20small% 20populations.htm

Answer:

To be able to comply with the journal policy and the ethical restrictions imposed by the ethics committee, we shared all the relevant results in a Supporting Information file. We uploaded the tables with the planning scores and their correlations with gray matter in the public repository “figshare” with the DOI: 10.6084/m9.figshare.13672873. All of the above does not show personal information of the participants.

However, the individual patients' data is confidential, and we are prevented from publicly sharing these data.

These are the answers to the points that deserve attention:

1. “The neuropsychological evaluation was rather limited in scope (only MoCA and SOC). This should be acknowledged as a limitation in the Discussion.”

Answer:

We agree with the reviewer’s suggestion. We have addressed this concerns in the potential limitations section of our study. 

3) In this study we used two tests to analyze the cognitive impairment in general and the planning deficits in particular. It is recommended that for future assessments of the global cognitive status, and the planning ability, more neuropsychological batteries should be used, allowing a more comprehensive evaluation in these patients.

(lines 287-291).

Moreover, we are currently monitoring our participant group with a more comprehensive cognitive battery to better characterize their cognitive profile.

2. The MoCA showed differences between groups in general cognitive ability. The authors could consider taking this into account as a covariate (maybe MoCA without executive component), so as to have a more "pure" estimate of planning ability as measured with SOC in association with the VBM measure.

Answer:

We thank the reviewer suggestion. In fact, we used the MoCA score as a covariate in the voxel-based morphometry analysis. However, we did not find any effect when it was included vs. when it was not included. This could be due to the fact that the MoCA performance was very heterogeneous among the patients, since some of them had larger deficits in the memory domain, others only in attention and others in language. The above explains why this variable did not pass the Shapiro-Wilk normality test (p <.004).

3. The relevance of striatum is briefly mentioned in the Discussion, but as frontostriatal involvement has been regarded to be crucial for planning performance, the authors should devote a few more sentences to the question how this relates to their results.

Answer: 

We agreed with the reviewer’s suggestion; therefore, we added the following paragraph in the Discussion section. 

Usually, cognitive impairments exhibited in HD patients are associated to caudate nucleus decrease due to the spiny neurons atrophy (33). This decrease seems to affect particularly the caudate nucleus during the clinical debut, specially compromising the prefrontal cortico-striatal loop (2, 31, 32). However, our study found that putamen deterioration in HD patients also contributes to the accuracy decrease to solve planning problems. This result suggests that, in addition to the caudate nucleus, the putamen degeneration could also compromise cognitive functions during the early clinical stage.

(lines 268-280). 

The requirements of the expert Reviewer are:

1. “Confusingly, the next sentence (lines 205-207) states: ‘No significant associations were found on the accuracy to solve all problems neither on the efficiency to solve “difficult” problems.’ What do they mean? What do I miss or misunderstand?”

Answer:

We are sorry for making this mistake in the drafting of the text. We have corrected the sentence as follows (and now is consistent with figure 2):

No significant correlations were found in the efficiency to solve all problems nor the accuracy to solve “difficult” problems.

(lines 204-205).

2. “The association between SOC test performance and loss of gray matter density appears to vindicate a role in early HD for the degeneration of posterior parts in what have traditionally been considered ‘frontal executive functions’. But the problem that should be recognized in this type of analysis is the widespread degeneration that may take place and the resulting collinearity of data with resulting overinterpretation of correlations. It would be nice if this issue would be addressed in the Discussion.” 

Answer:

We totally agree with the reviewer comment. We added this suggestion as one of the limitations in the Discussion section as mentioned earlier in the point number 2 of the Editor's comments:

5) Finally, even though we identified temporo-occipital degeneration associated with planning performance, we must be cautious with the interpretation of this result. The reported correlations passed the multiple comparison corrections, and each correlation showed different coefficients of determination, however, they could still have a collinearity problem, which is a main challenge to overcome in this type of study.(lines 294-299).

3. “In the materials and Methods section, first line, 22 HD mutation carriers and 20 healthy controls are mentioned. But Table 1 of Results, a control group of 22 (men:women = 9:13) is mentioned”.

Answer: 

We thank the reviewer for pointing out this mistake. We have corrected the healthy controls Men:Women ratio in Table 1.

4. How were symptomatic patients defined and selected? As having specific motor signs (the international definition) or of having early cognitive, behavioural or functional problems prior to the onset of motor signs? I assume the latter. The ‘Clinical testing’ section mentions only 5 patients with ‘abnormal movements’ and 2 patients with UHDRS motor score of 0. The authors should explain their patient selection in more detail.

Answer:

We rewrote the participants section to improve the description. The main criterion to determine the early clinical status of the subjects was the Total Functional Capacity scale from UHDRS:

Particularly, the symptomatic HD patients were defined by the scores obtained in the TFC scale, where a lower score reflects a reduced functional capacity. The TFC range is from 0 to 13 points, consequently, in our study we defined as early symptomatic HD to the subjects with scores higher than 7, considering the scale criteria (15). 

(lines 101-105)

5. In the Discussion the authors write: ‘our analysis found a significant correlation between the RACG with low efficiency planning, and RPCG with low accuracy planning; both frontal brain regions.’ (Lines 258 and 259) I guess they mean: both limbic regions?

Answer:

We agreed with the reviewer’s suggestion. We have changed this conceptualization:

…limbic brain regions.

(line 251).

---

## [Editor Report · Decision Letter 1]

12 Mar 2021

Planning deficits in Huntington's disease: a brain structural correlation by voxel-based morphometry

PONE-D-20-37811R1

Dear Dr. Gálvez Zúñiga,

We’re pleased to inform you that your manuscript has been judged scientifically suitable for publication and will be formally accepted for publication once it meets all outstanding technical requirements.

Kind regards,

Andre Aleman, PhD

Academic Editor

PLOS ONE
---

## [Editor Report · Acceptance letter]

16 Mar 2021

PONE-D-20-37811R1 

Planning deficits in Huntington's disease: a brain structural correlation by voxel-based morphometry 

Dear Dr. Galvez:

I'm pleased to inform you that your manuscript has been deemed suitable for publication in PLOS ONE. Congratulations! Your manuscript is now with our production department. 

Kind regards, 

on behalf of

Dr. Andre Aleman 

Academic Editor

PLOS ONE